# Evaluation of the medically necessary, time sensitive triage score during and beyond the local COVID-19 pandemic in the Gynaecologic Oncology Unit of a tertiary hospital in South Africa

**Adekunle Emmanuel Sajo**[1] *, **Arie Mouton**[1,2], **Gbenga Olorunfemi**[3], **Visser Cathy**[1], **Matthys Cornelis van Aardt**[1,2], **Greta Dreyer**[1,2]

1 Department of Obstetrics and Gynaecology, Gynaecologic Oncology Unit, University of Pretoria, Pretoria, South Africa, 2 Department of Obstetrics and Gynaecology, Gynaecologic Oncology Unit, Steve Biko Academic Hospital, Pretoria, South Africa, 3 Division of Epidemiology and Biostatistics, School of Public Health, University of Witwatersrand, Johannesburg, South Africa

* adekunlesajo@yahoo.com

## Abstract

### Objective

The main objective of this study was to evaluate the Medically Necessary Time Sensitive (MeNTS) scoring system in triaging gynaecologic oncologic surgery during and beyond the COVID-19 pandemic.

### Material and methods

This was a retrospective cross-sectional study including 209 patients who either had surgery (151) or surgery postponed (58) between the 26th March and 30th September 2020 in an academic hospital in South Africa. The MeNTS score was used to independently score each patient three times by two observers.

### Results

The mean age of the participants was 46.6 ± 15 years and the cumulative mean MeNTS score was 51.0 ± 5.1. Over two-thirds of the cases had surgery. There was no significant difference between the first and second observers' cumulative scores, 51.0 vs 51.1 (p 0.77). The cumulative score among those who had surgery was significantly lower than that for those whose surgeries were postponed, 49.8 vs 54.1 (p <0.0001). The intra-observer and inter-observer reliability were 0.78 and 0.74 respectively. After adjusting for confounding variables, those with low cumulative MeNTS scores were about 5 times more likely to have surgery than those with high scores (Adj. OR = 4.67, 95% CI: 1.92–11.4, *p* <0.001. Patients with malignant diagnosis were also 5 times more likely to be operated than those with benign diagnosis (Adj. OR = 5.03, 95% CI: 1.73–14.6, *p* <0.001. The area under the curve (AUC)

**Data Availability Statement:** All relevant data are within the manuscript and its supporting information files.

**Funding:** NO- There was no funding from any source for this research.

**Competing interests:** The authors have declared that no competing interest exists.

was 0.85 suggesting an excellent discriminatory power between those who were operated and those who were postponed.

## Conclusion

The study provided some insight into the potential usefulness of MeNTS score in prioritizing patients for surgery in gynaecologic oncologic sub-specialty. The score performed well across a range of gynaecologic conditions and procedures with good intra-observer and inter-observer consistency and reliability. This is a prioritization tool that is dynamically adaptable to accommodate changes in resources availability and operating theatre capacity.

## Introduction

In the year 2020, the world outlook and healthcare delivery were disrupted by the declaration of a pandemic, coronavirus 19 (COVID-19). Every aspect of our lives was affected and more especially the healthcare delivery.

The South African Government, just like most other countries across the world, declared a total lockdown on the 26th March 2020 with suspension of most non-urgent healthcare cases, including some oncologic services, due to lack of understanding of the dynamics of the pandemic. The outpatient clinics were suspended, the operating theatre capacity was significantly reduced to accommodate mostly emergency cases and there was redistribution of some surgical staff to areas of pressing need. The gynaecologic oncologic unit of our hospital was sparingly allocated theatre time initially at the peak of the pandemic to cater for the surgical needs of the cancer patients and those with debilitating benign cases on a case by case basis.

We were confronted with the ethical dilemma of balancing safety and equitable approach in objectively prioritizing patients for surgery within the context of limited theatre time and increased numbers of patients scheduled for surgery. Those that were operated were prioritized on a case-by-case approach as judged by the senior oncologists without any objective prioritization scale. We also followed the guidelines and recommendations of international gynaecologic societies to decide the best treatment modality and the radicality of procedures to minimize increased consumption of resources and prolonged hospital stay [1–4]. However, these recommendations were not based on any objective prioritization scale but on sound clinical judgement and consensus agreement which could be biased. Therefore, an objective tool that would incorporate the time sensitive nature of the disease as well as the operation, consumption of resources and the risk of transmission of COVID-19 to the surgical team and patients is required in surgical oncology practice as well as all other surgical specialties.

A scoring system deemed objective in prioritizing surgical patients was developed during the peak of the pandemic. This Medically Necessary, Time-Sensitive (MeNTS) score was developed by Prachand and colleagues at the University of Chicago, USA, based on expert opinions and consensus [5]. This tool integrates 21 variables generated from three factors namely procedure factor, disease factor and patient factor. Each variable is assigned a point from 1 to 5 with cumulative MeNTS score from the summation of the three factors ranging from 21 to 105. A higher cumulative score is associated with poorer perioperative outcomes, increased consumption of resources and an increased risk of transmission of COVID-19 to the healthcare personnel. A dynamic upper threshold score can be decided by the surgical team based on the availability of resources and operating theatre capacity. Above this threshold, cases can be

postponed as carrying out such procedure at that time may likely not be justified because of the attendant consequences that would impact resources consumption.

It was estimated that during the first global COVID-19 pandemic peak, a total of 28.4 million operations were cancelled or postponed and about 2.4 million surgical procedures were cancelled weekly, of which 8.2% were cancer related cases [6]. It was also estimated that about 45 weeks would be required to clear the backlog of surgeries resulting from the postponed or cancelled cases during this period [6]. Women with cancers are a particularly vulnerable group in Africa. Considering the fact that the surgical needs of most patients in Africa and other low-income countries are not sufficiently met, denying them access to healthcare during the pandemic would amount to trampling on their rights [7]. And the postponement of treatment of cancers would cause disease progression with great impact on the quality of life of affected persons and poor oncologic outcomes. It is also known that cancers that become upstaged during the waiting time would be more expensive to treat apart from the detrimental impact on the quality of life of the patients beyond six weeks of delay [8–10].

The MeNTS score has been found adaptable and useful in other specialties such as paediatric surgery, urology and cardiology by modification of the variables to arrive at a cumulative score based on needs [10–12]. Many times, oncology procedures are scheduled on a first come, first served basis and some cases are given priority after review on a case-by-case merit. This subjective system of prioritization is fraught with inconsistencies and bias, and has been associated with prolonged waiting time and patient dissatisfaction. It is therefore not certain if the utilization of MeNTS scoring tool in surgical oncology would help to reduce bias and ethical dilemma of prioritizing patients during and beyond the critical periods of resources shortage. We aimed to evaluate the MeNTS scoring tool in triaging gynaecologic oncologic surgery during and beyond the COVID-19 pandemic.

## Material and methods

### Study design and study population

This was a retrospective cross-sectional descriptive study carried out in the Gynaecologic Oncology Unit (GOU) of the department of Obstetrics and Gynaecology, Steve Biko Academic Hospital, Pretoria between the 26th March and 30th September 2020. The study population consisted 232 patients that either had surgery for gynaecologic conditions or had their surgery cancelled or postponed during the pandemic period. Participants were identified through the GOU's operation booking book and theatre book. Their hospital records (electronic and file records) were reviewed to allocate appropriate scores to the recruited participants.

This study was approved by The Research Ethics Committee, Faculty of Health Sciences, University of Pretoria with Reference number 279/2021 dated 18th June 2021 after the approval of the study protocol by the Master of Medicine (MMED) Committee, Faculty of Health Sciences, University of Pretoria. Patient data were first accessed on the 1st August 2021. The need for participant consent was waived by the Research Ethics Committee.

### Scoring

The MeNTS scoring system based on pre-operative characteristics as originally described by Prachands et al. [5] was used to score all the patients who had surgical procedures at the discretion of the team or on a first come first served basis and also those whose procedures were cancelled or postponed. Three major factors were considered in the MeNTS design: procedure factor (seven variables), disease factor (six variables) and patient factor (eight variables). Each of the variables has a 5-point scale and the cumulative scores range between 21 and 105 points.

Two independent observers were involved in the scoring of the participants. The principal investigator used the MeNTS tables to score each patient twice with an interval of at least two weeks. The second observer also scored each patient separately using the same scoring tool. Each patient therefore had three scores. Patients were anonymised by allocating study numbers to ensure each patient was scored correctly. Data sheet was used to capture the scores and other clinical characteristics before being transferred to Excel sheets for final analysis.

## Statistical analysis

The statistical analysis was conducted using Stata version 13 statistical package (Stata I/C, StataCorp LP, Texas, USA). Frequency and percentage, and graphs were produced to describe demographic and clinical characteristics of the participants. Normality was checked using the Skewness-Kurtosis test. The continuous variables that were normally distributed were presented as means ± standard deviation while non-normally distributed continuous variables were presented as median and interquartile range. The Student's t-test was used to compare the cumulative mean MeNTS scores across two categories of age group, surgery status and BMI group. Paired t-test was used to compare the cumulative mean MeNTS scores between the two observers. Mann-Whitney U test was used to compare the median score across CD4 count category and other non-normally distributed continuous variables. Multivariate tests of means analysis was done to compare the mean scores of procedures, disease and patient factors. One-way Analysis of Variance (ANOVA) or Kruskal Wallis test was used to compare the mean or median score across three or more categorical variables followed by the Bonferroni post hoc test. The association between categorical variables and categories of MeNTS score was assessed using the Pearson's Chi-square test or Fischer's exact test. Two-way ANOVA test was used to determine the interaction of diagnosis and operation types on the MeNTS scores. The test-retest (intra-observer) and inter-observer reliability were assessed using the Cronbach's Alpha. The univariable and multivariable logistic regression analysis were conducted between the independent variables and the surgery status category. Variables with p-value <0.2 were used in a stepwise regression model to build the final multivariable model. The AUC was determined to evaluate the discriminatory ability of the model. Comparison of different ROCs analysis was used to assess the reliability of the two models. The level of statistical significance was set at p-value <0.05, confidence interval of 95% and the two–tailed test of hypothesis was assumed.

## Results

We recruited 232 participants who were scheduled for operation between the 26[th] March and 30[th] September 2020, 23 (9.9%) participants had missing files and they were subsequently removed from the analysis. The mean age of participants was 46.6 ± 15 yrs. As indicated in Table 1, there was no statistically significant difference in the baseline characteristics between the participants that were operated and those whose surgeries were deferred, except in the disease types, parity and comorbidity categories (p <0.05). The proportion of participants with malignant diseases who were operated was significantly higher than those with malignant disease whose operations were postponed, 47.0% vs 12.1%.

The mean body mass index (BMI) was 28.9 ± 7.2 kg/m$^2$. A greater proportion (56%) of the participants was HIV negative, and among those who were HIV positive, 89% had CD4 count ≥200 cells/μL. The median CD4 count was 477 cells/μL (122–865) with a range of 16–1662 cells/μL. Majority of the patients, 141 (67.5%) had no comorbidity. When multivariate tests of means analysis was done to compare the mean scores of the three factors used to calculate the MeNTS score, there was a statistically significant difference in their means (p <0.0001) and

**Table 1. Comparison of the baseline demographic characteristics between those operated and not operated.**

| Characteristics | | Surgery | | P-value |
|---|---|---|---|---|
| | | Done, n (%) | Postponed, n (%) | |
| **Age (years) mean ±SD** | | **46.6 ± 15** | | |
| <50 | | 91 (60.3) | 38 (65.5) | 0.48[a] |
| ≥50 | | 60 (39.7) | 20 (34.5) | |
| **Parity** | | | | |
| 0 | | 29 (19.2) | 19 (32.8) | 0.013[a] |
| 1–4 | | 116 (76.8) | 33 (56.9) | |
| ≥5 | | 6 (4.0) | 6 (10.3) | |
| **BMI, kg/m2** | | **28.9 ± 7.2 kg/m$^2$** | | |
| <18.5 | | 5 (4.5) | 1 (16.7) | 0.53[b] |
| 18.5–24.9 | | 31 (27.9) | 1 (16.7) | |
| 25–29.9 | | 24 (21.6) | 1 (16.7) | |
| ≥30 | | 51 (46.0) | 3 (50.0) | |
| **Obesity** | **No** | 60 (54.0) | 3 (50.0) | 0.85[a] |
| | **Yes** | 51 (46.0) | 3 (50.0) | |
| **Comorbidity** | | | | |
| None | | 90 (59.6) | 51 (87.9) | <0.001[a] |
| Single | | 43 (28.5) | 6 (10.4) | |
| Multiple | | 18 (11.9) | 1 (1.7) | |
| **HIV status** | | | | |
| Negative | | 91 (60.3) | 26 (44.8) | 0.66[a] |
| Positive | | 54 (35.8) | 18 (31.0) | |
| Unknown | | 6 (3.9) | 14 (24.2) | |
| **Disease type** | | | | |
| Benign | | 54 (35.8) | 45 (77.6) | 0.001[a] |
| Premalignant | | 26 (17.2) | 6 (10.3) | |
| Malignant | | 71 (47.0) | 7 (12.1) | |
| **CD4 count, cells/µL** | | **477 cells/µL, IQR (122–865)** | | |
| <200 | | 8 (17.4) | 0 (0) | 0.33[b] |
| ≥200 | | 38 (82.6) | 11 (100.0) | |
| **Procedure Factor score** | | 21 ± 4.9 | 20.0 ± 3.9 | 0.14[^] |
| **Disease Factor score** | | 14 (12–21) | 25 (21–26) | <0.0001[c] |
| **Patient Factor score** | | 13 (10–14) | 11 (9–13) | 0.0006[c] |

n = number, unpaired t-test[^]

Mann-Whitney U test[c]

Chi squared[a]

Fisher's exact[b]

the significance was sustained when the means were compared across surgery status, disease and operation types. There was no statistically significant difference between the procedure factor average scores for operated cases and postponed cases, 21 vs 20 (p 0.14). The disease factor median score was significantly lower among cases that were operated as compared to the postponed cases, 14 vs 25 (p <0.0001), while the patient factor median score was higher among operated cases when compared with the postponed cases, 13 vs 11 (p 0.0006). The minimum cumulative mean MeNTS score was 34 while the maximum score was 64. The score below 55 was categorized as low score. The intermediate score ranged between 55 and 59 while scores above or equal to 60 were categorized as high cumulative score (Fig 1).

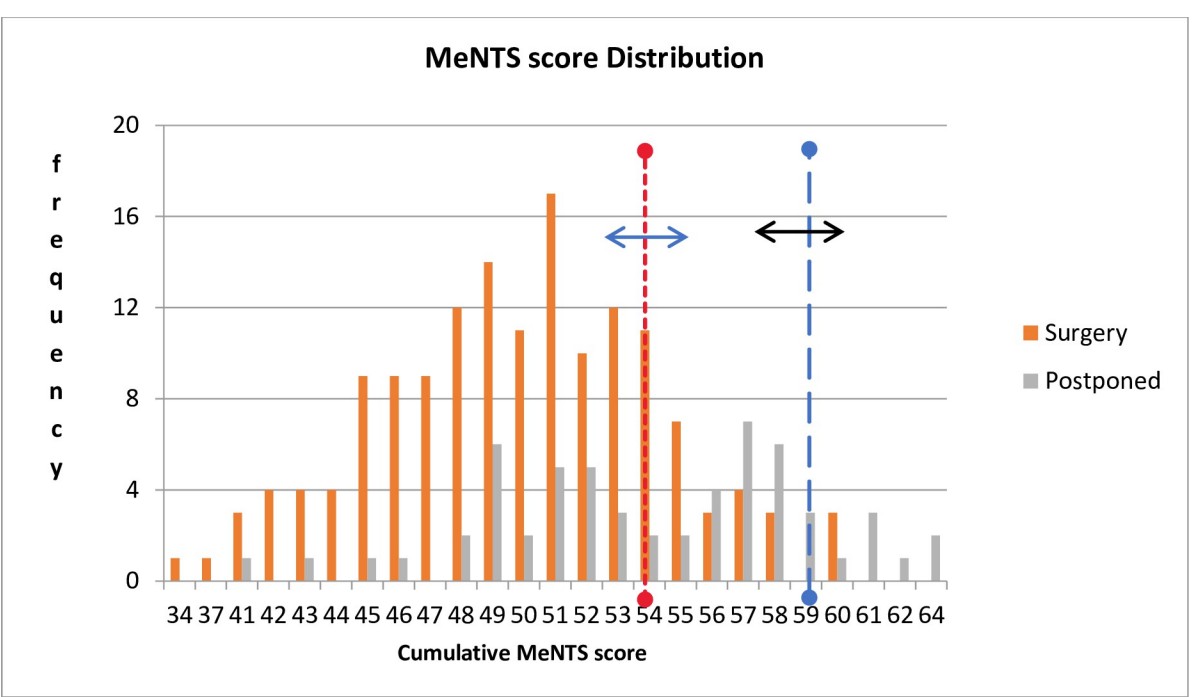

**Fig 1. The column graph showing the distribution of MeNTS scores.**

There was a statistically significant association between the categories of cumulative MeNTS score and whether surgery was done or postponed (p <0.001). Low cumulative MeNTS score as compared to high score, was significantly associated with surgery being done (p <0.001) as shown in Table 2.

A significant proportion (86.8%) of patients with low score was operated while equal proportion (50.0%) of patients with high scores either had surgery or had their surgery postponed. There was significant association between the disease types and category of MeNTS score (p <0.001). A greater percentage (96.2%) of those with malignant disease had low score when compared to those with benign disease (61.6%, p <0.001). There was no statistically significant difference in the cumulative mean MeNTS scores between the first and second observers' scores, 51.0 vs 51.1 (p 0.77). The cumulative mean MeNTS scores for those who were operated as compared to those who were not operated were significantly lower across the 2 observers, 49.8 vs 54.1 (p <0.0001) and 50.4 vs 52.8 (p 0.006) respectively. The Cronbach's Alpha coefficients for both intra-observer and inter-observer reliability were 0.78 and 0.74. There was statistically significant difference in the mean score across the disease type category, p < 0.0001. After post hoc test, the mean MeNTS score of those with malignant disease was significantly lower when compared to the mean scores of those with either premalignant or benign diseases (p <0.001). There was statistically significant relationship between the mean MeNTS score and the category of operation types, p <0.0001, as shown in Table 3.

After post hoc test, there was significant difference between those who had radical procedures when compared to those who had major procedures (48.1 vs 51.1, p 0.001). There was no significant difference in the mean score between those who had minor or major procedures. Among patients who were operated, there was no significant difference in the American Society of Anaesthesiologists Physical Status (ASA PS) classifications. There was no statistically significant interaction of both diagnosis types and operation types on the mean MeNTS score (p 0.22). However, simple main effect analysis showed that radical operation had a significant

**Table 2. Comparison of the cumulative mean scores between observers, across surgery status and other characteristics.**

| Characteristics | | Surgery | | | P value |
|---|---|---|---|---|---|
| | | Done n (%) | Postponed n (%) | | |
| **Three MeNTS score categories** | | | | | |
| Low (<55) | | 131 (86.8) | 29 (50.0) | | <0.001# |
| Intermediate (55–59) | | 17 (11.2) | 22 (37.9) | | |
| High (≥60) | | 3 (2.0) | 7 (12.1) | | |
| **Two categories** | | | | | |
| Low (<55) | | 131 (86.8) | 29 (50.0) | | <0.001# |
| High (≥55) | | 20 (13.2) | 29 (50.0) | | |
| **Observer 1** | | 49.8 ± 4.5 | 54.1 ± 5.1 | | <0.0001a |
| **Observer 2** | | 50.4 ± 5.6 | 52.8 ± 5.2 | | 0.006a |
| | | Disease type | | | |
| **Two MeNTS score categories** | | Benign | Premalignant | Malignant | |
| Low (<55) | | 61 (61.6) | 24 (75.0) | 75 (96.1) | <0.001# |
| High (≥55) | | 38 (38.4) | 8 (25.0) | 3 (3.9) | |
| Observer 1 | | 51.0 ± 5.1 | | | 0.77^ |
| Observer 2 | | 51.1 ± 5.6 | | | |
| BMI | Non-obese (63) | 50.2 ± 4.0 | | | 0.78a |
| | Obese (54) | 49.9 ± 4.8 | | | |
| Comorbidity | Single (49) | 50.7 ± 4.7 | | | 0.13^^ |
| | Multiple (19) | 53.2 ± 4.7 | | | |
| | None (141) | 50.8 ± 5.2 | | | |
| Age | <50 (129) | 50.6 ± 5.1 | | | 0.17a |
| | ≥50 (80) | 51.6 ± 4.9 | | | |
| CD4 | <200 (8) | 51 (47–53) | | | 0.96c |
| | ≥200 (49) | 51 (48–53) | | | |

Student's t test[a]

paired t test[^]

Mann-Whitney U test[c]ANOVA[^^]

Chi-square[#]

effect on the cumulative mean MeNTS score than other operation types when malignant diagnosis type was considered.

Univariate regression analysis showed that patients with a low MeNTS score were about seven times more likely to have surgery than those with high score (OR = 6.55, 95% CI: 3.26–13.16, p <0.001) as shown in Table 4.

Patients with malignant diagnosis were eight times more likely to be operated than those with benign diagnosis (OR = 8.45, 95% CI: 3.54–20.2, p <0.001). After adjusting for confounding variables such as diagnosis types, parity, HIV status and comorbidity, the odds of having surgery done among women with low MeNTS score was about five times higher than the odds in women with high cumulative MeNTS score (≥55) (Adj. odds = 4.67, 95% CI: 1.92–11.4, p 0.001). The odds of having surgery done was five times higher among women with malignant diagnosis than the odds among women with benign diagnosis (Adj. odds = 5.03, 95% CI: 1.73–14.6, p 0.003). Both observer models AUC excellently (85% vs 84%) discriminated those that had surgery from those who had their surgery postponed, after correcting for confounding variables (Fig 2).

This suggests there was no difference in the inter-observer relationship in the MeNTS scoring between the 2 observers.

**Table 3. Comparison of the cumulative mean MeNTS scores among categorical variables.**

| Characteristics | frequency | Mean ± SD | P value |
|---|---|---|---|
| **Disease type** | | | |
| Benign | 99 | 52.4 ± 5.6 | <0.0001[a] |
| Premalignant | 32 | 52.7 ± 3.1 | |
| Malignant | 78 | 48.5 ± 3.9 | |
| **Operation type** | | | |
| Diagnostic | 11 | 44.9 ± 5.5 | <0.0001[a] |
| Minor | 7 | 51.4 ± 2.4 | |
| Major | 88 | 51.1 ± 4.4 | |
| Radical | 45 | 48.1 ± 3.3 | |
| **Parity (median, IQR)** | | | |
| 0 | 48 | 51 (47–55.5) | 0.028[b] |
| 1–4 | 149 | 51 (48–54) | |
| ≥5 | 12 | 55 (52.5–57.5) | |
| **HIV Status** | | | |
| Negative | 117 | 50.6 ± 5.2 | 0.28[a] |
| Positive | 72 | 51.1 ± 4.6 | |
| Unknown | 20 | 52.6 ± 5.9 | |
| **BMI** | | **(median, IQR)** | |
| <18.5 | 6 | 52.5 (50–54) | 0.75[a] |
| 18.5–24.9 | 32 | 49.5 (47–52) | |
| 25–29.9 | 25 | 51 (48–53) | |
| ≥30 | 54 | 49.5 (47–53) | |
| **ASA** | | **(median, IQR)** | |
| 1 (Normal) | 33 | 49 (46–53) | 0.37[b] |
| 2 (mild/moderate disease) | 77 | 51 (47–53) | |
| 3 (Severe disease) | 32 | 49 (46–51) | |

ASA = American Society of Anaesthesiologists Physical Status

ANOVA[a]

Kruskal-Wallis[b]

## Discussion

This study sought to evaluate the MeNTS score in prioritizing gynaecologic oncologic patients for surgery by conducting a retrospective cross-sectional study among patients who were operated and those whose surgeries were postponed during the period of national COVID-19 lockdown in South Africa. To the best of our knowledge, this was the first study that would evaluate MeNTS score among gynaecologic oncologic patients. Overall, our study suggested that the MeNTS score had an excellent likelihood of discriminating between cases that were prioritized for surgery and those that were deferred. We also found that cases that were operated had significant lower cumulative mean MeNTS score than those whose surgeries were postponed. Patients with a malignant gynaecologic diagnosis had a significantly lower mean score than those with either benign or premalignant diagnosis. There was also no statistically significant difference when patients were scored at different times by the same observer or by different individual, suggesting good intra-observer and inter-observer reliability. Therefore, our findings suggest MeNTS score could be a decision-making tool in prioritizing patients for surgery in gynaecologic oncology. These findings align with those reported by Prachand and colleagues in prioritizing surgical patients in the time of resources redistribution [5].

**Table 4. Univariate and multivariate logistic regression for the predictors of surgery.**

| Variables | OR | 95% CI | p value | Adj OR | 95% CI | p value |
|---|---|---|---|---|---|---|
| **MeNTS score** | | | | | | |
| High (≥55) | 1.00 | reference | reference | 1.00 | reference | reference |
| Low (<55) | 6.55 | 3.26–13.16 | <0.001** | 4.67 | 1.92–11.4 | <0.001** |
| **Diagnosis type** | | | | | | |
| Benign | 1.00 | Reference | reference | 1.00 | reference | reference |
| Premalignant | 3.61 | 1.37–9.54 | 0.01 | 4.02 | 1.08–15.03 | 0.038 |
| Malignant | 8.45 | 3.54–20.2 | <0.001** | 5.03 | 1.73–14.6 | 0.003** |
| **Parity** | | | | | | |
| 0 | 1.00 | reference | reference | 1.00 | reference | reference |
| 1–4 | 2.30 | 1.15–4.62 | 0.02** | 1.13 | 0.44–2.89 | 0.80 |
| >4 | 0.66 | 0.18–2.34 | 0.51 | 0.10 | 0.01–0.73 | 0.023** |
| **HIV status** | | | | | | |
| Negative | 1.00 | reference | reference | 1.00 | reference | reference |
| Positive | 0.86 | 0.43–1.71 | 0.66 | 0.78 | 0.30–2.04 | 0.61 |
| Unknown | 0.12 | 0.04–0.35 | <0.001** | 0.09 | 0.02–0.39 | <0.001** |
| **Comorbidity** | | | | | | |
| Multiple | 1.00 | reference | reference | 1.00 | reference | reference |
| None | 0.10 | 0.01–0.76 | 0.026** | 0.01 | 0.001–0.22 | 0.003** |
| Single | 0.40 | 0.05–3.55 | 0.41 | 0.05 | 0.003–0.92 | 0.044** |
| **Age group** | | | | | | |
| <50 | 1.00 | reference | reference | | | |
| ≥50 | 1.25 | 0.67–2.36 | 0.49 | | | |
| **BMI (kg/m$^2$)** | | | | | | |
| <25 (Underweight/Normal) | 1.00 | reference | reference | | | |
| 25–29 (Over weight) | 1.33 | 0.11–15.53 | 0.82 | | | |
| ≥30 (Obese) | 0.94 | 0.15–5.94 | 0.95 | | | |

OR = Odd ratio, Adj

OR = adjusted odd ratio

CI: 95% Confidence interval

** significance

Saleeby and co-workers modified the MeNTS score to prioritize patients with benign gynaecologic conditions to different surgical modalities ranging from minimal access to open surgery [13]. The median score in their study was 33 which was lower than 51 reported in our study. The majority of the studies where the MeNTS score was modified, reported lower cumulative mean scores than our finding [11–14]. This might be due to the differences in the weight allocated to each variable in the modified scores. However, they all found the operated cases to have lower scores than the postponed cases, except in the finding of Coello and co-worker where there was no difference in the scores between the operated and the deferred cases [15]. This further suggests the triaging capability of the MeNTS score as a surgical prioritization tool. The second published study that evaluated the MeNTS score in gynaecology was also done on patients with benign diagnosis [16]. Their reported mean Gyn-MeNTS score of 58 and 59 across three observers' ratings were higher than the total mean scores of 51.0 and 51.1 for the first and second observers in our study respectively. However, they reported an excellent AUC of 0.89 in discriminating those operated from the non-operated groups. This was similar to the AUCs of 0.85 and 0.84 for the two models in our study. This implies further

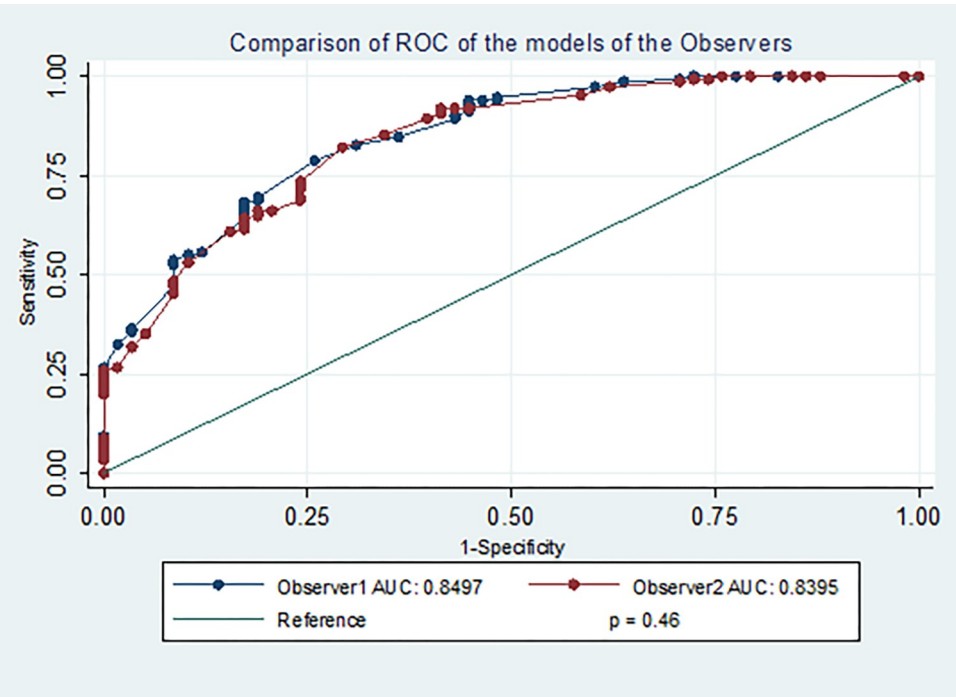

**Fig 2. Comparison of ROC of the two models generated from observer 1 and observer 2.**

that the MeNTS score might find some usefulness in gynaecology in prioritizing either benign or malignant diseases for surgery. In contrast to the 67% of the cases which were given high priority for surgery using the Gyn-MeNTS, more than 80% of our cases were classified as low score [16]. This might be because all the Gyn-MeNTS cases were benign conditions as compared to only 47% benign cases in our study.

Medical comorbidity is an important consideration in planning patients for surgery, as this could impact their outcomes negatively. It is also an important consideration in gynaecologic oncology to assist in deciding on the radicality of the primary operation or consideration for other treatment options. This was an important variable in the patient factor arm of the MeNTS score calculation. However, the severity of the comorbidity was only assessed by the number of medications taken by the patients rather than how impactful the condition is on their functionality. Only one-third of our participants had one or multiple comorbidities such as cardiovascular disease, lung disease or diabetes mellitus as compared to the 79% of the participants reported by the Saleeby et al. [13]. We did not find any statistically significant difference in the cumulative mean score between those with or without comorbidity. This might be because some of the comorbidities were important considerations in calculating the cumulative MeNTS score.

There were three factors considered in the calculation of the MeNTS score: procedure, disease and patient factors. We reported a significant lower mean score of 14 in the disease factor among those operated as compared to a mean score of 25 among those who had postponed surgery. This was higher than the median disease factor scores among the operated and non-operated groups (9.5 vs 11) reported by Coello et al, but was similar to what Prachand and co-workers described in the original MeNTS scoring [5, 15]. This suggests a significant proportion of those operated had diseases that might impact negatively on their survival and quality of life if surgical intervention was delayed or they had diseases with no known effective

conservative care. This further emphasizes the urgency in treatment of oncological patients as subsequent care might be significantly hampered if treatment is delayed. The benign cases that were operated during the restrictive period were those likely to have debilitating effect on the quality of life of the patients. Only about a third of the benign cases scheduled during the study period were operated, probably due to the debilitating impacts of their conditions. Our decisions to operate benign conditions during this restrictive period was in line with the recommendations of the Society of Gynecologic Oncology (SGO) and other international societies to prioritize easily metastasizing or aggressive tumours, cancer surgery with curative intent and operation of benign diseases that could have debilitating effects on the patients [1–5, 17, 18]. The lower mean disease factor score among the operated group in this study might be due to the greater proportion of patients with malignant diagnosis. We also found a significantly higher mean patient factor of 13 among those operated as compared to mean score of 11 in the non-operated group suggesting an increased risk of COVID-19 illness and increased risk of postoperative respiratory impairment requiring mechanical ventilation. However, only one of our patients required mechanical ventilation in critical care unit postoperatively and one acquired severe COVID-19 infection postoperatively while on admission, culminating in her death.

The cut-off MeNTS score of 55 to discriminate between low and high score in our study was similar to the one described in the original score by Prachand and colleagues [5]. There was a good distribution of the score across the operated and non-operated groups in terms of age, disease types and surgery types. Similar normal distribution was reported by Saleeby et al. and Marfori et al. in their studies respectively [13, 16]. The intermediate score group (score 55–59) forms the flexible range of the upper and lower threshold MeNTS score to determine who will be operated or postponed. This shows the dynamism of the MeNTS score in assisting response to the changes in healthcare capacities, resources availability and risk tolerance of each hospital. It would allow for the preservation of theatre capacity for highly prioritized cases and emergencies. Patients with benign diseases whose surgeries are more likely to be postponed because of their above-threshold scores, can be allowed to be operated by dynamically adjusting the threshold. In this way, patients can be accommodated in order to avoid debility with the aim of improving their quality of life. This flexibility makes MeNTS score a valuable prioritization tool in gynaecologic oncology. Generally, lower cumulative scores indicate that surgical procedures are at lower risk for increased resource consumption, while higher scores indicate surgery that are not really time sensitive or there is higher risk for resources consumption if the procedure is carried out.

Our analysis showed good intra-observer internal consistency and reliability suggesting an expectation of similar consistent score when the same case is scored by the same individual at different times and conditions. This is the only study to evaluate intra-observer reliability of the MeNTS score. We also found an inter-observer reliability of 0.74, similar to the strong interrater reliability reported by the previously published studies on MeNTS score among gynaecologic patients [13, 16]. The AUC of 0.85 vs 0.84 from the two models in our study showed an excellent discriminating power of the MeNTS score for predicting who had surgery or not. The non-significant difference of the AUCs of the two models also suggests an excellent inter-observer reliability of the MeNTS score [19]. In a study by Dincer and co-workers that first evaluated the postoperative complication using the MeNTS score, there was low ability (AUC = 0.69) of the score in discriminating between those with no to mild postoperative complication and those with moderate to severe complication [20]. They also showed that patients with high scores were the ones with severe complications. This supports the decision to postpone cases with high scores as they could result in high consumption of the healthcare resources when complications occur. Our study was however not designed to evaluate postoperative complications.

The higher odds of patients with low MeNTS score having surgery done as compared to the odds of surgery postponement suggests that the lower the score, the more likely it is to proceed with the intended surgery. This further suggests the prioritization usefulness of the original MeNTS score and its other modifications described in other surgical specialties such as Urology, Orthopaedics and Paediatric surgery [5, 11, 14–16]. However, Coello and colleagues did not find any significant difference in the median score between the operated and deferred cases in the application of their modified MeNTS score in prioritizing urologic patients for surgical intervention [15].

The retrospective design of this study posed some limitations as it was difficult to draw accurate validation of the triaging tool. The bias with such design was however reduced by the scoring of each participant by two different observers. Also, the significant differences observed in both disease and patient factors among those operated and those whose surgery was postponed could have impacted significantly on the overall MeNTS score. This could be explained by the significant higher numbers of oncological pathology and morbidity among the operated group.

## Conclusion

This study provided some insights into the potential usefulness of the MeNTS score in prioritizing patients for surgery in gynaecologic oncologic specialty even though it is yet to be validated. The MeNTS score performed well across a range of gynaecologic conditions and procedures with good intra-observer and inter-observer consistency and reliability. This suggests MeNTS score as a prioritization tool that is dynamically adaptable to accommodate changes in resource availability and hospital theatre capacity. At the time of the current report, most hospitals and operating theatre capacities have slowly returned to the pre-pandemic status. Even though there is limited number of debilitating COVID-19 cases across the country, the post COVID-19 economic meltdown that currently plagues many African countries still hamper full resources allocation to healthcare system. Therefore, an objective triaging tool like the MeNTS score is still relevant in prioritizing patients for surgery beyond the COVID-19 pandemic. However, further validation of the scoring system is required.

## Supporting information

**S1 Dataset.**
(XLS)

## Acknowledgments

We acknowledged Steve Biko Academic Hospital Medical Health Records and Gynaecologic clinic staff who assisted in retrieving the patient medical records and my colleague who assisted in scoring the participants as the second observer.

The result of this study was presented at the Annual Global Meeting of International Gynecologic Cancer Society (IGCS) Conference in New York, USA (September 29-October 1, 2022).

## Author Contributions

**Conceptualization:** Adekunle Emmanuel Sajo, Arie Mouton, Gbenga Olorunfemi, Visser Cathy, Greta Dreyer.

**Data curation:** Adekunle Emmanuel Sajo, Visser Cathy.

**Formal analysis:** Adekunle Emmanuel Sajo, Gbenga Olorunfemi.

**Methodology:** Adekunle Emmanuel Sajo, Gbenga Olorunfemi, Greta Dreyer.

**Supervision:** Arie Mouton, Matthys Cornelis van Aardt, Greta Dreyer.

**Validation:** Matthys Cornelis van Aardt, Greta Dreyer.

**Visualization:** Greta Dreyer.

**Writing – original draft:** Adekunle Emmanuel Sajo, Greta Dreyer.

**Writing – review & editing:** Adekunle Emmanuel Sajo, Visser Cathy, Matthys Cornelis van Aardt, Greta Dreyer.

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
