## [Decision Letter · Decision Letter 0]

27 Feb 2023

PONE-D-22-35746Evaluation of the Medically Necessary, Time Sensitive Triage Score During and Beyond the Local COVID-19 Pandemic in the Gynaecologic Oncology Unit of a Tertiary Hospital in South AfricaPLOS ONE

Dear Dr. Sajo ,

Thank you for submitting your manuscript to PLOS ONE. After careful consideration, we feel that it has merit but does not fully meet PLOS ONE’s publication criteria as it currently stands. Therefore, we invite you to submit a revised version of the manuscript that addresses the points raised during the review process.

Please submit your revised manuscript by Apr 13 2023 11:59PM If you will need more time than this to complete your revisions, please reply to this message or contact the journal office at plosone@plos.org. Please include the following items when submitting your revised manuscript:A rebuttal letter that responds to each point raised by the academic editor and reviewer(s). You should upload this letter as a separate file labeled 'Response to Reviewers'.A marked-up copy of your manuscript that highlights changes made to the original version. You should upload this as a separate file labeled 'Revised Manuscript with Track Changes'.An unmarked version of your revised paper without tracked changes. You should upload this as a separate file labeled 'Manuscript'.If applicable, we recommend that you deposit your laboratory protocols in protocols.io to enhance the reproducibility of your results. Protocols.io assigns your protocol its own identifier (DOI) so that it can be cited independently in the future. For instructions see: https://journals.plos.org/plosone/s/submission-guidelines#loc-laboratory-protocols. Additionally, PLOS ONE offers an option for publishing peer-reviewed Lab Protocol articles, which describe protocols hosted on protocols.io. Read more information on sharing protocols at https://plos.org/protocols?utm_medium=editorial-email&utm_source=authorletters&utm_campaign=protocols.

We look forward to receiving your revised manuscript.

Kind regards,

Stefano Turi

Academic Editor

PLOS ONE

Journal Requirements:

  "NO- There was no funding from any source for this research"

   "No authors have competing interest"

6. Please ensure that you include a title page within your main document. You should list all authors and all affiliations as per our author instructions and clearly indicate the corresponding author.

Reviewers' comments:

Reviewer's Responses to Questions

**Comments to the Author**

1. Is the manuscript technically sound, and do the data support the conclusions?

Reviewer #1: Yes

Reviewer #2: Yes

2. Has the statistical analysis been performed appropriately and rigorously? 

Reviewer #1: Yes

Reviewer #2: Yes

3. Have the authors made all data underlying the findings in their manuscript fully available?

Reviewer #1: Yes

Reviewer #2: Yes

4. Is the manuscript presented in an intelligible fashion and written in standard English?

Reviewer #1: Yes

Reviewer #2: Yes

5. Review Comments to the Author

Reviewer #1: Congratulacions to he authors for their interesting study reporting usefulness of the MENTS score in prioritizing patients for surgery in gynaecologic oncologic speacialty.

As the authors of the study clearly demonstrate, the MENTS score is a good tool t0 prioritize resources in times when they are restricted in pandemic situations. The results of this paper confirm for the first time that it is applicable to more specialized units within the gynecology specialty, such as oncology.

In general, I agree with the introduction, with the description of material and statistical method, presentation of results and conclusions.

However, I want to make the following comment that I think should be put as a limitation to the study in the final part of the discussion. The authors find a clear difference in the total MENTS score, and therefore conclude that it is a valid tool to discriminate the patients who will undergo surgery; however, if we analyze each factor of the MENTS score separately, we can observe that there are very significant differences in disease and patient factor, and none in the procedure. These very marked differences in these two factors can weigh heavily on the overall MENTS score. The explanation that could be given is that the oncological pathology and morbidity were significantly higher in the operated group This situacion is commonly seen in specialized oncological units.

Except for this comment, which should be cited as a limitation of the study, it is shown that the MENTS score is a useful tool in the selection and prioritization of patients who must be operated in times of resource restriction. For all of the above, I believe that this paper meets the characteristics to be published in PLOS-ONE.

Reviewer #2: There are a few typographical errors which should be corrected. The use of the word non-emergent is inappropriate and should be replaced with non-urgent. The authors evaluated the MeNTS in Gynecologic Oncologic settings but not describe it. It was done by Prachand et al.They should delete this from the abstract and body. The scoring system could be applicable in other resource constrained situations but should be validated and re-evaluated on a continuous basis.

6. PLOS authors have the option to publish the peer review history of their article (what does this mean?). If published, this will include your full peer review and any attached files.

Reviewer #1: No

Reviewer #2: **Yes: **Professor Isaac Adewole

---

## [Author Response · Author response to Decision Letter 0]

10 Mar 2023

Thank you for the thorough review of our manuscript and for the consideration of this study in your reputable journal. 

We have critically examined the comments and suggestions and each of the raised issues is addressed below;

Editor’s comments

1. Comment: Please ensure that your manuscript meets PLOS ONE's style requirements, including those for file naming on PLOS ONE’s style

Response: The manuscript has been reviewed to reflect PLOS ONE’s style

2. Comment: Please provide additional details regarding participant consent

Response: The need for participant consent was waived by the Research Ethics Committee as it was a retrospective study. Patients medical records were retrieved but all the data retrieved were fully anonymized.

3. Comment: on financial disclosure

Response: The authors did not receive specific funding for this work. This statement has been included in the cover letter

4. Comment: on competing interests

Response: The authors have declared that no competing interest exists. This statement has been included in the cover letter and in the online submission form. 

5. Comment: In your Data Availability statement, you have not specified where the minimal data set underlying the results described in your manuscript can be found

Response: The minimal data set used to arrive at the results in this study is submitted as supporting document with this manuscript. Any other enquiry can be directed to the corresponding author. This statement has also been included in the cover letter. 

6. Comment: Please ensure that you include a title page within your main document. You should list all authors and all affiliations as per our author instructions and clearly indicate the corresponding author

Response: A title page with the list of the authors and their affiliations as per PLOS ONE’s author instructions has been included within the main document. 

7. Comment: Please review your reference list to ensure that it is complete and correct. If you have cited papers that have been retracted, please include the rationale for doing so in the manuscript text, or remove these references and replace them with relevant current references

Response: The authors went through the reference list, the list is complete and correct, except for the name of the third author in reference 15 which was wrongly spelt as “de le Cruz” instead of “de la Cruz”. This has been corrected in the main document. 

None of the reference list has been retracted to the best of the authors’ search. 

Reviewers’ comments

8. First reviewer’s comment: I want to make the following comment that I think should be put as a limitation to the study in the final part of the discussion. The authors find a clear difference in the total MENTS score, and therefore conclude that it is a valid tool to discriminate the patients who will undergo surgery; however, if we analyse each factor of the MENTS score separately, we can observe that there are very significant differences in disease and patient factor, and none in the procedure. These very marked differences in these two factors can weigh heavily on the overall MENTS score. The explanation that could be given is that the oncological pathology and morbidity were significantly higher in the operated group This situation is commonly seen in specialized oncological units.

Response: These statements have been added to the final part of the discussion: “The retrospective design of this study posed some limitations as it was difficult to draw accurate validation of the triaging tool. The bias with this design was however reduced by scoring each participant by two different observers. Also, the significant differences observed in both the disease and patient factors among those operated and those whose surgery was postponed could have impacted significantly on the overall MeNTS score. This could be explained by the significant higher numbers of oncological pathology and morbidity among the operated group”. 

9. Second Reviewer’s comment: There are a few typographical errors which should be corrected. The use of the word non-emergent is inappropriate and should be replaced with non-urgent. The authors evaluated the MeNTS in Gynaecologic Oncologic settings but not describe it. It was done by Prachand et al. They should delete this from the abstract and body.

Response: The write up has been thoroughly perused and corrections of the few typographical errors have been made. The word non-emergent has been replaced with non-urgent. The use of the word “describe” has been replaced with “evaluate” in the abstract and body of the manuscript as suggested by the reviewer

All other corrections are shown in track changes in the main manuscript.

---

## [Editor Report · Decision Letter 1]

27 Mar 2023

Evaluation of the Medically Necessary, Time Sensitive triage score during and beyond the local COVID-19 pandemic in the Gynaecologic Oncology Unit of a Tertiary Hospital in South Africa

PONE-D-22-35746R1

Dear Dr. Adekunle Emmanuel Sajo,

We’re pleased to inform you that your manuscript has been judged scientifically suitable for publication and will be formally accepted for publication once it meets all outstanding technical requirements.

Kind regards,

Stefano Turi

Academic Editor

PLOS ONE

---

## [Editor Report · Acceptance letter]

4 Apr 2023

PONE-D-22-35746R1 

Evaluation of the Medically Necessary, Time Sensitive triage score during and beyond the local COVID-19 pandemic in the Gynaecologic Oncology Unit of a Tertiary Hospital in South Africa 

Dear Dr. Sajo:

I'm pleased to inform you that your manuscript has been deemed suitable for publication in PLOS ONE. Congratulations! Your manuscript is now with our production department. 

Kind regards, 

on behalf of

Dr. Stefano Turi 

Academic Editor

PLOS ONE